A comparative analysis of the complete chloroplast genome sequences of four peanut botanical varieties

http://orcid.org/0000-0002-0500-3900 Wang Juan
Li Chunjuan
Yan Caixia
Zhao Xiaobo
Shan Shihua shansh1971@163.com
Shandong Peanut Research Institute , Qingdao , China
Ribeiro-Barros Ana
Electronic publication date: 2018 Jul 31
Publication date: 2018
Volume: 6
Electronic Location ID: e5349
Received 2018 Jan 29; Accepted 2018 Jul 10
Copyright: © 2018 Wang et al.
Copyright year: 2018
Copyright holder: Wang et al.
License: This is an open access article distributed under the terms of the Creative Commons Attribution License, which permits unrestricted use, distribution, reproduction and adaptation in any medium and for any purpose provided that it is properly attributed. For attribution, the original author(s), title, publication source (PeerJ) and either DOI or URL of the article must be cited.
License URL: https://creativecommons.org/licenses/by/4.0/

Keywords: Arachis hypogaea, Comparative cp genomes, Genetic structure, Genetic variation

Funding: Natural Science Foundation of Shandong Province ZR2017BC082 Central Guidance for Local Science and Technology; Taishan Scholars Project ts201712080 Agricultural Science and Technological Innovation Project of Shandong Academy of Agricultural Science CXGC2018E21 This work was supported by the Natural Science Foundation of Shandong Province (ZR2017BC082); the Specific Funds of the Central Guidance for Local Science and Technology; Taishan Scholars Project (ts201712080); Agricultural Science and Technological Innovation Project of Shandong Academy of Agricultural Science (CXGC2018E21). The funders had no role in study design, data collection and analysis, decision to publish, or preparation of the manuscript.

==============================
Background

Arachis hypogaea L. is an economically important oilseed crop worldwide comprising six botanical varieties. In this work, we characterized the chloroplast (cp) genome sequences of the four widely distributed peanut varieties.

Methods

The cp genome data of these four botanical varieties (var. hypogaea, var. hirsuta, var. fastigiata, and var. vulgaris) were obtained by next-generation sequencing. These high-throughput sequencing reads were then assembled, annotated, and comparatively analyzed.

Results

The total cp genome lengths of the studied A. hypogaea varieties were 156,354 bp (var. hypogaea), 156,878 bp (var. hirsuta), 156,718 bp (var. fastigiata), and 156,399 bp (var. vulgaris). Comparative analysis of theses cp genome sequences revealed that their gene content, gene order, and GC content were highly conserved, with only a total of 46 single nucleotide polymorphisms and 26 insertions/deletions identified. Most of the variations were restricted to non-coding sequences, especially, the trnI-GAU intron region was detected to be highly variable and will be useful for future evolutionary studies.

Discussion

The four cp genome sequences acquired here will provide valuable genetic resources for distinguishing A. hypogaea botanical varieties and determining their evolutionary relationship.

Introduction

Cultivated peanut (Arachis hypogaea L.) is one of the most important oilseed crops that is mainly planted in China, India, USA, and Argentina (Hammons, 1994; Grabiele et al., 2012; Bertioli et al., 2016). Based on morphological (Gibbons, Bunting & Smartt, 1972; Krapovickas & Rigoni, 1960; Krapovickas & Gregory, 2007) and molecular (Gepts, 1993; He & Prakash, 2001) evidences, six botanical varieties of A. hypogaea have been identified: var. hypogaea, var. hirsuta, var. fastigiata, var. vulgaris (Gibbons, Bunting & Smartt, 1972), as well as var. aequatoriana and var. peruviana with the last two being region specific.

In land plants, the cp genome is circular and has a large single copy (LSC) region and a small single copy (SSC) region that are separated by a pair of inverted repeat (IR) regions (Raubeson & Jansen, 2005). The major role of the chloroplast (cp) is to conduct photosynthesis; additionally, it is involved in the biosynthesis of fatty acids, vitamins, pigments, and amino acids (Prabhudas et al., 2016). Different from nuclear sequence, the cp DNA has several advantages, including low-recombination, haploid ploidy, and maternal inheritance, making cp DNA an ideal tool for evolutionary studies (Birky, 2001; Wu & Ge, 2012). For example, with the help of genetic markers that include two non-coding cpDNA regions (trnTR-trnS and trnT-trnY), Grabiele et al. (2012) found that the six peanut botanical varieties were very likely to have a single genetic origin, however, the fine evolutionary relationship between these varieties remains to be resolved.

The rapid progress of high-throughput sequencing technology development has greatly facilitated the acquisition of cp genome data, which are not only powerful for reconstructing interspecific phylogeny (Jansen et al., 2007; Parks, Cronn & Liston, 2009; Moore et al., 2010), but are also helpful for investigating genome dynamic at the subspecies level. For instance, Zhao et al. (2015) compared the cp genomes of four Chinese Panax ginseng strains and suggested that their genome dynamic was under selective pressure.

Although there are six botanical varieties within A. hypogaea that differ at both the morphological and molecular levels (Ferguson, Bramel & Chandra, 2004), only very limited A. hypogaea cp genome data are currently available (Prabhudas et al., 2016; Choi & Choi, 2017). Here, we acquired and examined the complete cp genome nucleotide sequences of the four main peanut botanical varieties, providing valuable genetic resources for further evolutionary studies.

Materials and Methods

DNA extraction and sequencing

Four representative A. hypogaea varieties (var. hypogaea, var. hirsuta, var. fastigiata, and var. vulgaris) were collected from Shandong Peanut Research Institute, Qingdao, China. China has become the largest producer of cultivated peanut in the world (Yu, 2008), and these four main botanical varieties have been cultivated in China for more than 500 years. The seedlings were grown using hydroponic methods. The cp DNA was isolated from fresh leaves (>5 g) of 3- to 4-week-old seedlings using the Plant Chloroplast DNAOUT Kit (Bjbalb, Beijing, China). The quality of cp DNA samples was checked by agarose gel electrophoresis with Super GelRed (US Everbright Inc., Suzhou, China). Libraries with an average length of 350 bp were constructed using the NexteraXT DNA Library Preparation Kit (Illumina, Shanghai, China). The quality of the libraries was checked by GeneRead DNA QuantiMIZE Assay Kit (Qiagen, Duesseldorf, Germany). Sequencing was performed on the Illumina HiSeq Xten platform (Illumina, Shanghai, China), and the average length of the generated reads was 150 bp.

Data assembly and annotation

The quality of the raw paired-end reads was assessed by FastQC v0.11.3 (Andrews, 2014). All raw data for four A. hypogaea varieties were filtered based on the following rules: (1) adapter trimming; (2) quality control; each read has <5% unidentified nucleotides and >50% of its bases with a quality value of >20. This filtration was carried out using Cutadapt v1.7.1 (Martin, 2011). The high-quality data were then assembled into contigs using the de novo assembler SPAdes v3.9.0 (Nurk et al., 2013), and these contigs were further assembled into complete cp genome using NOVOPlasty (Dierckxsens, Mardulyn & Smits, 2017). The assembled data were checked against the published complete cp genome of A. hypogaea (GenBank accession no. KX257487, Prabhudas et al., 2016). The cp genes were annotated using the DOGMA tool with default parameters (Wyman, Jansen & Boore, 2004). The cp genome images were drawn with OGDraw v1.2 (Lohse, Drechsel & Bock, 2007).

Variation detection and evolutionary relationship analysis

Multiple sequence alignment was generated using VISTA and Mauve v2.3.1 software (Frazer et al., 2004; Darling, Mau & Perna, 2010) and was checked manually when necessary. All alignments were visualized using the VISTA viewer program (Mayor et al., 2000). Single nucleotide polymorphisms (SNPs) were identified by Mauve v2.3.1. The insertions/deletions (InDels) were retrieved from the sequence alignments using the mVISTA package. An InDels image including 10 bp up- and downstream was then generated. Simple sequence repeats (SSRs) were isolated from all filtered InDels. Repeat sequences with repeating units of 2–6 bp that repeated no fewer than three times were considered as SSRs.

The genetic relationship of the four peanut cp genomes together with two available peanut cp genome sequences (GenBank accession no. KX257487 and KJ468094; Prabhudas et al., 2016; Choi & Choi, 2017) were examined by constructing a minimum evolutionary (ME) tree using MEGA v6 with default parameters (Tamura et al., 2011). The cp genome sequences from four other related species (Robinia pseudoacacia, Ceratonia siliqua, Leucaena trichandra, and Senna tora) of Fabaceae were used as outgroups (CSI-BLAST E-value < 10−6).

Results

Assembly and validation of cp genomes

More than 1 GB raw sequencing data per sample was generated from high-throughput sequencing. After cleaning and trimming, 22,511,400 (var. vulgaris) to 62,087,400 (var. hirsuta) paired-end reads were acquired, which were then mapped separately to the reference cp genome, attaining coverage amounts of 143× to 396×. After performing de novo and reference-guided assembly with minor modifications, we acquired four complete cp genome sequences for A. hypogaea var. hypogaea, var. hirsuta, var. fastigiata, and var. vulgaris (Fig. 1; Table 1).

Figure 1 Gene map of the A. hypogaea chloroplast genomes.

Genes shown outside the outer circle are transcribed clockwise and those inside are transcribed counterclockwise. Genes belonging to different functional groups are color-coded. Dashed area in the inner circle indicates the GC content of the chloroplast genome.

Table 1 Genes identified in the chloroplast genome of peanut.

Category for genes	Group of genes	Name of genes	
Self-replication	tRNA genes	rrn5, rrn4.5, rrn16, rrn23	
rRNA genes	*trnA-UGC, trnC-GCA, trnD-GUC, trnE-UUC, trnF-GAA, trnG-GCC, *trnG-UCC, trnH-GUG, trnI-CAU, *trnI-GAU, *trnK-UUU, trnL-CAA, *trnL-UAA, trnL-UAG, trnfM-CAU, trnM-CAU, trnN-GUU, trnP-UGG, trnQ-UUG, trnR-ACG, trnR-UCU, trnS-GCU, trnS-GGA, trnS-UGA, trnT-GGU, trnT-UGU, trnV-GAC, *trnV-UAC, trnW-CCA, trnY-GUA	
Small subunit of ribosome	rps2, rps3, rps4, rps7, rps8, rps11, *rps12, rps14, rps15, *rps16, rps18, rps19	
Large subunit of ribosome	rpl2, rpl14, *rpl16, rpl20, rpl22, rpl23, rpl32, rpl33, rpl36	
DNA dependent RNA polymerase	rpoA, rpoB, *rpoC1, rpoC2	
Genes for photosynthesis	Subunits of NADH-dehydrogenase	*ndhA, *ndhB, ndhC, ndhD, ndhE, ndhF, ndhG, ndhH, ndhI, ndhJ, ndhK	
Subunits of photosystem I	psaA, psaB, psaC, psaI, psaJ	
Subunits of photosystem II	psbA, psbB, psbC, psbD, psbE, psbF, psbH, psbI, psbJ, psbK, psbL, psbN, psbT, psbZ	
Subunits of cytochrome b/f complex	petA, *petB, *petD, petG, petL, petN	
Subunits of ATP synthase	atpA, atpB, atpE, *atpF, atpH, atpI	
Large subunit of rubisco	rbcL	
Other genes	Maturase	matK	
Protease	*clpP	
Envelope membrane protein	cemA	
Subunit of acetyl-CoA-carboxylase	accD	
C-type cytochrome synthesis gene	ccsA	
Genes of unknown function	Open reading frames (ORF, ycf)	ycf1, ycf2, *ycf3, ycf4	
Note:

Intron-containing genes are marked by asterisks (*).

For each of the assembled cp genome sequences, a .sqn file that was generated by the Sequin software (https://www.ncbi.nlm.nih.gov/projects/Sequin/), submitted to GenBank and acquired the following accession numbers: MG814006 for var. fastigiata, MG814007 for var. hirsuta, MG814008 for var. hypogaea, and MG814009 for var. vulgaris. Users can download the data for research purposes only when referencing this paper.

Genetic structure of the peanut cp genome

These four acquired peanut cp genomes were found to have the classical quadripartite structure of land plant cp genomes that comprises a LSC, a SSC, and two IR (A/B) regions. The sequence lengths among the four cp genomes ranged from 156,354 to 156,878 bp. The size varied from 85,900 bp (var. hirsuta) to 86,196 bp (var. fastigiata) in the LSC region, from 18,796 bp (var. hypogaea, var. hirsuta, and var. vulgaris) to 18,874 bp (var. fastigiata) in the SSC region and from 25,806 bp (var. hypogaea) to 26,091 bp (var. hirsuta) in the IR (A/B) region (Table 1). A total of 110 genes were identified from the cp genome: four ribosomal RNA (rRNA) genes, 76 protein-coding genes, and 30 transfer RNA (tRNA) genes (Table 2). Among the 110 identified genes, six protein-coding genes, six tRNA genes, and four rRNA genes were distributed in the IR (A/B) regions. The cp genome consisted of 55.66% coding regions and 44.34% non-coding regions including both intergenic spacers and introns. The overall GC content of the cp genomic sequences was 36.3–36.4%, and the GC contents of the LSC, SSC, and IR (A/B) regions were 33.8%, 30.2–30.3%, and 42.8–42.9%, respectively (Table 2).

Table 2 Details of the complete chloroplast genomes of four peanut botanical varieties.

	AHL	AHZ	AHP	AHD	
Matched reads (bp)	62,087,400	22,511,400	61,928,100	34,570,200	
Genome size (bp)	156,878	156,399	156,354	156,718	
Mean coverage (×)	395.77	143.94	396.08	220.59	
LSC length (bp)	85,900	85,955	85,946	86,196	
SSC length (bp)	18,796	18,796	18,796	18,874	
IR length (bp)	26,091	25,824	25,806	25,824	
LSC GC content (%)	33.8	33.8	33.8	33.8	
SSC GC content (%)	42.9	42.9	42.9	42.9	
IR GC content (%)	30.3	30.3	30.3	30.2	
GC content (%)	36.4	36.4	36.4	36.3	
Total number of genes	110	110	110	110	
Protein coding genes	76	76	76	76	
rRNA	4	4	4	4	
tRNA	30	30	30	30	

Variation among the cp genomes

Among the four acquired peanut cp genome sequences, there was no difference at the junction positions (Fig. 2). A total of 46 SNPs were found within the quadripartite structural region. VISTA-based identity plots illustrated the hotspot regions of genetic variation among the cp genomes (Fig. 3). As expected, non-coding sequences exhibited more variation than the coding sequences, and greater amounts of substitutions were found in the trnI-GAU intron (25 SNPs) and the ycf3-psaA spacer (eight SNPs) regions. The only identified non-synonymous was located within the psaA gene. The hydrophobic amino acid Tyr in var. hypogaea, var. fastigiata, and var. vulgaris was replaced by the hydrophilic amino acid Asn in var. hirsuta.

Figure 2 The comparison of the LSC, IR, and SSC border regions among the four peanut chloroplast genomes.

Figure 3 Visualization of alignment of the peanut chloroplast genome sequences.

Genome regions are color-coded as protein coding, rRNA coding, tRNA coding, or conserved noncoding sequences (CNS). The x-axis represents the coordinate in the chloroplast genome. Annotated genes are displayed along the top. The sequences similarity of the aligned regions is shown as horizontal bars indicating the average percent identity between 50% and 100%.

A total of 26 InDels were identified: 13 were located in spacers, nine were in introns, and four were in genes; 15 were in the LSC region, two were in the SSC region, and nine were in IR (A /B) regions (Fig. S1). Among these InDels, large InDels (>50 bp) were found in the psbK–trnQ intergenic spacer, the trnL intron, and ycf1. Meanwhile, we identified six SSR regions (sequence identity >90%): four A stretches and one T stretches ranging from 7 to 16 bp, as well as one with a CTAG repeat motif. No C or G stretches were identified. Moreover, InDels in the ycf1 and the ycf2 regions represent frameshift mutations: the 63 bp-insertion at the end of the ycf1 gene led to a longer amino acid sequence in var. fastigiata, while a 18 bp-deletion was found in the middle of IR (A/B) ycf2 gene regions in var. hypogaea.

Genetic relationship analysis

Due to low genetic diversity, the whole cp genome sequences were used to construct an evolutionary tree based on ME algorithms. The results showed that these peanut cp genomes clustered into a monophyletic branch, while the four outgroup species were clustered into another branch. Among the six analyzed peanut cp genomes, var. hirsuta is relatively different from the rest and constitute a basal clade (Fig. 4). The high-support values (>99%) were shown above the nodes.

Figure 4 The evolutionary relationship among four cultivated peanuts and the related species of Fabaceae constructed by NJ analyses.

Numbers above node are bootstrap support values.

Discussion

The cp is an important plant cell organelle (Alberts et al., 2002). The cp genome usually lacks recombination and is maternally inherited and is therefore very useful for distinguishing taxa and inferring evolutionary relationships. Here, we have studied the cp genomes of cultivated peanut (A. hypogaea) that is an economically important oilseed crop worldwide. A. hypogaea comprised six varieties that differ at both the morphological and molecular levels (Ferguson, Bramel & Chandra, 2004). So far, only very limited A. hypogaea cp genome data are available (Prabhudas et al., 2016).

In the present study, we acquired and closely examined the whole cp genome sequences of four main peanut varieties. We found that the overall cp genome structures of the four botanical varieties were the same and displayed the classical quadripartite structure of land plant cp genome (Raubeson & Jansen, 2005). No definitive genomic rearrangements or gene inversions were found among the four peanut cp genomes. The sequence variation among the four peanut cp genomes was also relatively limited, and most of them were restricted to the non-coding regions, especially the trnI-GAU intron exhibited an outstanding level of variation (25 out of the entire 46 identified SNPs), suggesting that the rapidly evolving nature of this intron. This trnI-GAU intron has therefore a great potential for developing molecular markers that could be used in future phylogenetic studies.

In addition, a minimum-evolution tree of the four acquired peanut cp genomes together with two earlier published peanut cp genomes has been constructed to speculate their evolutionary relationships. Our result showed that the six investigated peanut cp genomes form a monophyletic branch, and this agrees with earlier studies (Grabiele et al., 2012). In addition, our result also revealed that among the six studied peanut cp genomes, var. hirsuta was relatively more distantly related to the others and may constitute a basal branch, which was in line with the previous reports (Duan et al., 1995; Ferguson, Bramel & Chandra, 2004). Consistent with its suggested relationship between var. hirsuta and the other studied peanut varieties, var. hirsuta appeared to be the peanut variety found within the archeological remains along the Pacific coast of Perú (Bonavia) that may be the region of origin of cultivated peanut (Simpson, 2001; Stalker, 2017).

Conclusion

With the help of high-throughput sequencing technology, we revealed the complete cp genomes of four main peanut botanical varieties. The gene contents and gene orders of the cp genomes were highly conserved. The trnI-GAU intron region was considered to be rapid-evolving region that could potentially serve as molecular markers in phylogenetic studies. This study will provide valuable cp genomic resources for future exploitation.

Supplemental Information

Supplemental Information 1 Fig. S1. InDels alignment and with 10 bp-upstream and downstream sequences.

Click here for additional data file.

Supplemental Information 2 Table S1. SNP distribution of peanut cp genome.

Click here for additional data file.

Supplemental Information 3 Assembled file of var. hypogaea.

Click here for additional data file.

Supplemental Information 4 Assembled file of var. hirsuta.

Click here for additional data file.

Supplemental Information 5 Assembled file of var. vulgaris.

Click here for additional data file.

Supplemental Information 6 Assembled file of var. fastigiata.

Click here for additional data file.

The authors would like to acknowledge Dr. Yuan Li from Lund University of Sweden and Dr. Dachuan Shi from Qingdao Academy of Agricultural Science of China for their excellent advice on this paper.

Additional Information and Declarations

Competing Interests

Author Contributions

DNA Deposition

Data Availability

The authors declare that they have no competing interests.

Juan Wang conceived and designed the experiments, performed the experiments, analyzed the data, contributed reagents/materials/analysis tools, prepared figures and/or tables, authored or reviewed drafts of the paper, approved the final draft.

Chunjuan Li performed the experiments, analyzed the data, contributed reagents/materials/analysis tools, prepared figures and/or tables, approved the final draft.

Caixia Yan performed the experiments, authored or reviewed drafts of the paper, approved the final draft.

Xiaobo Zhao performed the experiments, authored or reviewed drafts of the paper, approved the final draft.

Shihua Shan conceived and designed the experiments, analyzed the data, authored or reviewed drafts of the paper, approved the final draft.

The following information was supplied regarding the deposition of DNA sequences:

The assembled complete cp genome sequence is available at NCBI GenBank: Accession numbers MG814006 (AHD); MG814007 (AHL); MG814008 (AHP); MG814009 (AHZ). Raw sequence are available on NCBI SRA (BioProject Accession No. PRJNA481236).

The following information was supplied regarding data availability:

The raw data are provided in the Supplemental Files.

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
