# Peer review of "A comparative analysis of the complete chloroplast genome sequences of four peanut botanical varieties"

_PeerJ, doi:10.7717/peerj.5349_

## Round 0.1 · original submission · Major Revisions

Dear Dr J Wang,

After receiving the evaluations of four reviewers, it is my opinion that although the data is interesting and relevant, the MS should be considerably improved before publication. Please see this as a positive criticism towards the quality of your paper. I will be happy to receive the revised version.

Best regards
Ana Ribeiro-Barros

Reviewer 1 ·

Basic reporting

The structure of the article conforms with PeerJ standards, the article has many minor grammatical inconsistencies. In figure 2, the y- axis title has a typographical error.

Experimental design

The mentioned methodology is sound and backs the results aptly, however I would suggest to include the tool used for genome assembly in lines 101-102 and line 145-153.

Validity of the findings

The authors have provided enough rational explanation for backing their study and it is commendable.

Additional comments

The outcome of the work is novel and I think it would attract researchers to look into this aspect of chloroplast genome variation among the variants of the same species. I would recommend the article for publications after necessary revisions have been implemented.

·

Basic reporting

A lot of good data have been generated, but the results need to be revised thoroughly to derive the complete meaning of the data.

Language needs a major revision

Experimental design

Appropriate

Validity of the findings

Good

Additional comments

Major concerns:
I strongly recommend to check the language before publishing this work. Better to get a review from native English speakers or Professionals.
Comments:
Page 1: Background: The first sentence is not meaningful. Hence need to be clearly written.
Method: The sentence “According to the alignment sequences, the genome-wide genetic variations (SNPs and InDels) were developed”

Discrepancy in usage of terms “cultivars vs botanical types”; “Genome vs cp Genome”; “America vs American” need to be checked in the entire manuscript. Need to maintain uniformity.

Line#17: Abstract is not written properly. It is copied from the Background, method, discussion part as such.

Line#39: Keywords can be more specific.
Line#40: Short title can be changed.
Specific doubts:
Line#78: Are the nomenclatures AHP, AHL, AHD and AHZ are globally accepted?

Line#82: “genetic relationships based on four peanut cultivars” Are they of different botanical types? Need to specify.
Line#88: What are the genotype names of these botanical types?
Line#114: What are IR and SC region. Explain the details
Line#115: How many relative species- indicate the plant species used in Phylogenetic analysis.
Line#129: Cannot trace accession numbers at NCBI. Will it be accessible to public after publication of this MS?
Line#132: “Size and gene content of the peanut genome” Is it done for whole genome or cp genome? In the paragraph need to explain the IR, LSC and SSC region. The abbreviations should be expanded at the first instance.
Line#138-140: The sentence is not clear.
Line#145: Explain about DNA flexibility. Mention the methods to calculate this in Methods section
Line#162: Need to explain about the criteria used to identify InDels.
Line#173: Is this six or four cp genome sequences?
In discussions, the relation between different botanical types has not been discussed but it is indicated in the title of the Manuscript.

·

Basic reporting

While this article is written in passable English, it is riddled with grammatical errors. I don't think that it should be the job of a peer reviewer to copy-edit the manuscript, but I would highly recommend that the authors seek the services of a copy editor to correct the grammar and English usage before resubmitting the manuscript.

Technically speaking, this article is thorough in providing background and context for the study, but possibly it has too much background information. While some basic information about the cultivation history of the species is useful, perhaps it could be limited to a single paragraph. Similarly, the background information about chloroplast sequencing and phylogenetics could be simplified to perhaps a single paragraph.

The data presented is fairly standard for a paper on chloroplast sequencing and phylogenetics. I don't understand the meaning of the analysis of DNA flexibility in this context; I've never seen this issue addressed in a phylogenetics paper and the authors provide no citations for what this means and why it's relevant to the study.

I could not find the sequences on Genbank, so I can't comment on the quality of the annotations.

Experimental design

I am somewhat confused as to what the purpose of this study was: was it to clarify the phylogenetic positions of the four landraces, or was it merely to provide evidence about the plastome sequences of them?

If the purpose was the first, I don't think that it is sufficient to use a single sample of each landrace and use only one linkage group (the maternally-inherited plastome). I'd like to see a study that uses these reference plastomes and resequences many samples of each landrace and uses more rigorous phylogenetic analyses to at least determine with confidence the plastome's phylogenetic history of the landraces.

If the purpose was the second, I guess I can't really argue with the methods used. I'd still remove the phylogenetic results and discussion, for the reasons I stated above, but I don't have any particular complaints about the methods used to sequence four chloroplast genomes. I have never seen a DNA flexibility analysis in a paper before (and the authors provided no citations for it), so I cannot judge how well that particular component was performed.

Validity of the findings

While it is always useful to add more reference sequences to Genbank, I can't tell if I think that that feat alone is enough to justify a paper. If the authors want to use this data to write a paper about the phylogenetics of landraces in Arachis, I think that the data presented here is a good start towards that analysis, but I don't think that the research and analysis presented here suffices for that purpose. I wouldn't trust a phylogenetic analysis using only a single sample each for an intraspecies study.

If the standard for PeerJ is merely that the information presented is technically accurate and provides some new benefit to the literature, a stripped-down version of this paper without the phylogenetic analyses would be sufficient, but I think the paper would be greatly improved by doing more robust phylogenetic analyses with more sampling.

Reviewer 4 ·

Basic reporting

The manuscript provides new data on the chloroplast complete sequences of four peanut varieties (six accessions). The results are original and add structural knowledge to the huge genomic resources that are being provided for peanut and its wild progenitors. However, there are several items to be improved before the manuscript can be considered for publication.

1. The English language should be improved in all the extension of the manuscript to ensure that an international audience can clearly understand the text. Some examples where already marked.

2. REFERENCES
Should be corrected and complete. Examples are:

Titles of the papers, with and withot capitals in all the words.
266 Gepts P. 1993. The Use of Molecular and Biochemical Markers in Crop Evolution Studies.
267 Evolutionary Biology-new York 27:51-94.
268 Gibbons RW, Bunting AH, and Smartt J. 1972. The classification of varieties of groundnut
269 (Arachis hypogaea L.). Euphytica 21:78-85.

Incomplete citations
289 Krapovickas et al. 1960. revista de investigaciones agricolas 14(2): 197-228.
Krapovickas, A. & Rigoni, V. A., 1960. La nomenclatura de las subespecies y variedades de Arachis hypogaea L. Revta Invest. agric. B. Aires. 14: 198–228.
305 Martin M. 2011. Cutadapt removes adapter sequences from high-throughput sequencing reads.
306 2011 17. 10.14806/ej.17.1.200
307 pp. 10-12


3. The title is so ambitious because the variability and phylogenetic data is inferred only from six accessions considering only four taxonomic varieties. Indeed, to understand the genetic variation in cultivated peanut, with more than 10.000 accessions in international gene banks, and to make a confident phylogenetic analysis a larger sample is needed.

Experimental design

.The sample used to infer the variability and to make a phylogenetic analysis is not appropriate, because of the huge number of peanut local races distributed worldwide, and kept in th einternational gene banks.

Validity of the findings

4. The phylogenetic results are discussed citing statements that are false. For instance,
218 AHL is the most similar to wild species morphologically (Krapovickas et al. 1960).
This statement is not true. In the world monograph of Arachis (Krapovickas & Gregory 1994, English translation edited in 2006), as well as in the reference cited by the authors it is clearly established that A. hypogaea hypogaea var hypogaea presents the characters most similar to those found in wild species. Simpson 1996, also stated that the var. hypogaea presents the most ancestral characters.
219 importantly, AHL is regarded as the earliest peanut cultivar that was domesticated in the South
220 American based on the historical record.
This sentence is very confusing and the statement is also not true. A. hypogaea var. hirsuta, is the variety found in the most ancient archeological remains, in the Pacific coast of Perú (Bonavia). However there is no evidence that this variety is the earliest domesticated. It is probably not, since the remains were found very far from where the ancestral wild species are distributed (Simpson et al 2001, Stalker et al. 2017).

---

## Round 0.2 · Minor Revisions

Although the new version has improved considerably, it is my opinion that it still deserves minor revisions. Particularly I woud like to highlight:

- Language and text editing: should be done to improve phrasing and formatting;

- Discussion: the rationale and conclusion drawn from the DNA flexibility analysis should be either addressed or otherwise removed.

Looking forward to receive the new version,
Sincerely
Ana Ribeiro-Barros

·

Basic reporting

The English in the manuscript is actually worse than it was in the last revision. I strongly recommend that the authors find a copy editor who is fluent in English to revise the manuscript: not only are there problems with the phrasing, but there are instances throughout where words are run together ("minimumevolutionary" on line 103, "Fabaceaeserved" on line 102), places where there are multiple spaces for no reason, repeated words (line 218), extra punctuation, etc. These sorts of mistakes are not related to the scientific merit of the paper, but demonstrate a carelessness in the revision process.

The discussion is improved from the previous version: I think that the authors clearly articulate why their limited phylogenetic analysis is still useful in supporting a previous hypothesized relationship within the varieties. There is no mention at all in the discussion of the rationale and conclusion drawn from the DNA flexibility analysis; either it should be discussed and justified in the discussion or it should be cut out entirely.

Experimental design

It seems like the authors merely mean to present some new plastome sequences within botanical varieties of A. hypogaea; if this is their purpose, it is well-explained and the paper does address this topic. The methods are clear enough to replicate the study.

Validity of the findings

I still don't understand the point of the flexibility analysis; while the authors address it in the rebuttal, none of that is present in the revised manuscript.

Reviewer 4 ·

Basic reporting

The authors attended all the major points requeted.

Experimental design

The authors attended all the major points requeted.

Validity of the findings

The authors attended all the major points requeted.

Additional comments

The authors attended all the major points requeted.

---

## Round 0.3 · Minor Revisions

Dear Dr Juan,

Thank you for your re-submission. Although this version could be acceptable for publication, I still think that the Abstract can be slightly improved to better describe your work. I introduced my suggestions in the attached MS. Please let me know if you agree with my suggestions.

Thanks for your understanding,

Ana Ribeiro-Barros

---

## Round 0.4 · accepted · Accept

Dear Dr Wang,

I am pleased to inform you that your MS is now accepted for publication. Thank you for your contribution and efforts to meet the quality standards of PeerJ.

Sincerely,
Ana I. Ribeiro-Barros

#